# The rs16969968 Tobacco Smoking-Related Single-Nucleotide Variant Is Associated with Clinical Markers in Patients with Severe COVID-19

**DOI:** 10.3390/ijms24129811

**Published:** 2023-06-06

**Authors:** Daniela Valencia-Pérez Rea, Ramcés Falfán-Valencia, Ingrid Fricke-Galindo, Ivette Buendía-Roldán, Leslie Chávez-Galán, Karol J. Nava-Quiroz, Jesús Alanis-Ponce, Gloria Pérez-Rubio

**Affiliations:** 1HLA Laboratory, Instituto Nacional de Enfermedades Respiratorias Ismael Cosío Villegas, Mexico City 14080, Mexico; daniela.980211@gmail.com (D.V.-P.R.); dcb_rfalfanv@hotmail.com (R.F.-V.); ingrid_fg@yahoo.com.mx (I.F.-G.); krolnava@hotmail.com (K.J.N.-Q.); 14alanisponce@gmail.com (J.A.-P.); 2Translational Research Laboratory on Aging and Pulmonary Fibrosis, Instituto Nacional de Enfermedades Respiratorias Ismael Cosío Villegas, Mexico City 14080, Mexico; ivettebu@yahoo.com.mx; 3Laboratory of Integrative Immunology, Instituto Nacional de Enfermedades Respiratorias Ismael Cosío Villegas, Mexico City 14080, Mexico; lchavezgalan@gmail.com

**Keywords:** COVID-19, *CHRNA5*, tobacco smoking, *ADAM33*, single nucleotide variant

## Abstract

Tobacco smoking is the leading risk factor for many respiratory diseases. Several genes are associated with nicotine addiction, such as *CHRNA5* and *ADAM33*. This research aims to evaluate the association of the polymorphisms rs16969968 (*CHRNA5*) and rs3918396 (*ADAM33*) in patients who developed severe COVID-19. We included 917 COVID-19 patients hospitalized with critical disease and oxygenation impairment. They were divided into two groups, tobacco-smoking (*n* = 257) and non-smoker (*n* = 660) patients. The genotype and allele frequencies of two single nucleotide variants, the rs16969968 (*CHRNA5*) and rs3918396 (*ADAM33*), were evaluated. The rs3918396 in *ADAM33* does not show a significative association. We analyzed the study population according to the rs16969968 genotype (GA + AA, *n* = 180, and GG, *n* = 737). The erythrocyte sedimentation rate (ESR) shows statistical differences; the GA + AA group had higher values than the GG group (*p* = 0.038, 32 vs. 26 mm/h, respectively). The smoking patients and GA or AA genotype carriers had a high positive correlation (*p* < 0.001, rho = 0.753) between fibrinogen and C-reactive protein. COVID-19 patients and smokers carriers of one or two copies of the risk allele (rs16969968/A) have high ESR and a positive correlation between fibrinogen and C-reactive protein.

## 1. Introduction

The symptomatology caused by SARS-CoV-2 infection is diverse, ranging from some individuals being asymptomatic to others developing acute respiratory distress syndrome (ARDS) and even requiring invasive mechanical ventilation (IMV). Several risk factors are associated with COVID-19 severity, like being older than 60 years old, male sex, having previous coexisting diseases, mainly systemic arterial hypertension, obesity, and being a smoker [1].

Tobacco smoking is one of the principal worldwide causes of morbidity and mortality [2,3]; it has been associated with the development of cardiovascular diseases [2,4], cancer [3,5], infection susceptibility [6,7], ARDS [3], and chronic lung diseases [3,5,6,7]. In the pandemic context of COVID-19, current smokers have an increased risk of presenting the worst outcomes [8]; in the airway, a higher expression of angiotensin-converting enzyme 2 (ACE2) [3] in smokers has been found, which is the receptor through the virus is internalized to the cells [9].

The susceptibility to the infection or disease severity in complex and multifactorial pathologies such as COVID-19, many factors are implicated, like environment, genetics, comorbidities, or addiction to certain drugs like nicotine [10]. Several genes are associated with nicotine addiction, such as *CHRNA5*, which codifies a subunit of the nicotine receptor of acetylcholine (nAchR) [4,11,12]. These proteins produce channels in the neurons, bronchial epithelial cells, and neuroendocrine lung cells [4,11]. A single nucleotide variant (SNV) rs16969968 (A allele) has been associated with nicotine addiction, lung cancer [4,7,11], schizophrenia [11], and chronic obstructive pulmonary disease (COPD) [4,11,13]. In bronchial cells, the neuronal acetylcholine receptor subunit alpha-5 protein modulates its adhesion and motility and regulates the expression of *p63*, a potential oncogene [14].

Another gene of interest but less studied is *ADAM33* which encodes a disintegrin and metalloproteinase domain-containing protein 33; this protein family is related to cellular adhesion, signaling, and proteolysis; variants in this gene have been principally associated with a higher risk for developing asthma, bronchial hyperreactivity, and COPD [15,16].

This research evaluated the association of the polymorphisms rs16969968 (*CHRNA5*) and rs3918396 (*ADAM33*) in patients who developed severe COVID-19. Both SNVs have been associated with COPD, asthma, and lung cancer.

## 2. Results

Genotype and allele analysis was made comparing smokers vs. non-smokers patients; the codominant model shows that does not exist an association between rs16969968 (*CHRNA5*) and rs3918396 (*ADAM33*) and COVID-19 (Table 1).

Next, we analyzed the study population according to the genotype of the rs16969968 (GA + AA *n* = 180, and GG *n* = 737); in both groups, there were men predominance (>65%), and age did not show statistical differences. In the comparison of body mass index (BMI), days since symptom onset, days of hospitalization, IMV requirement, and the number of deaths were not statistically different between groups; there was a tendency (*p* = 0.055) in PaO_2_/FiO_2_ ratio to be lower in the group of GA + AA. The groups also had a similar frequency of the most common comorbidities (Table 2). Obesity was the comorbidity more frequent (>40%) in the population, independently of the genotype. We made this analysis with rs3918396 (*ADAM33*); however, it did not show a significant association.

More frequent symptomatology (>40%) were fever, cough, dyspnea, myalgia, arthralgia, and headache. Thoracic pain was lower in the GA + AA group compared to GG (*p* = 0.047) (Table 3).

At admission time, the erythrocyte sedimentation rate (ESR) shows statistical differences; the GA + AA group had higher values than the GG group (*p* = 0.038, 32 vs. 26 mm/h, respectively) (Table 4).

We collected results of laboratory tests from the clinical record, including complete blood count, blood chemistry, and coagulation tests between the 7th–10th day of hospitalization; no showed statistical differences in the comparison GA + AA vs. GG (Appendix A).

We made the comparison intragroup of the clinical laboratory parameters at admission to the hospital and in the follow-up (7–10 days of hospitalization). Patients with genotypes GA or AA had a statically significant reduction (*p* < 0.05) in the lymphocyte count (2.0 vs. 0.8 × 103/μL), LDH (393 vs. 315 UI/L), CRP (12.0 vs. 7.6 mg/dL), fibrinogen (707 vs. 593 mg/dL), and ferritin (990 vs. 897 ng/mL).

The GG group presented statically significant changes (*p* < 0.05) in lymphocytes (2.1 vs. 0.8 × 103/μL), platelets (258 vs. 282 × 103/μL), LDH (387.5 vs. 325 UI/L), CRP (12.3 vs. 6.9 mg/dL), fibrinogen (666 vs. 609 mg/dL), and procalcitonin (0.21 vs. 0.13 ng/mL) (Appendix A).

The survival curve does not show differences in patients with GG or GA + AA (*p* = 0.5) (Figure 1).

For variables not correlated with each other at admission in the hospital, we made principal components analysis (PCA), patients included in the study were similar independently of genotype in rs16969968 (Figure 2).

Smokers’ patients were divided according to tobacco index (TI) in light smokers (TI < 20 packages-year, LS-COV, *n* = 178) and heavy smokers IT ≥ 20 packages-year (HS-COV, *n* = 56). Allele A (rs16969968) in *CHRNA5* was associated (*p* = 0.046) to risk (OR = 1.87; CI 95%, 1.05–3.34) in the HS-COV group. The genotype AA had an association significative to risk in the same group of patients (*p* = 0.034, OR = 7.23; CI 95%, 1.27–40.95), and the recessive model confirmed the significative association with the genotype of risk (AA) (*p* = 0.030, OR = 6.05, CI 95%, 1.40–25.88) in smokers patients with severe COVID-19 (Table 5). Variant rs3918396 (*ADAM33*) does not show a significative association.

Spearman’s analysis in patients with severe COVID-19 and smokers does not show a correlation between variables analyzed with genotype GG (rs16969968); however, the GA + AA group had a high positive correlation (*p* < 0.001, rho = 0.753) between fibrinogen and C-reactive protein. Regardless of the rs16969968 polymorphism (GG vs. GA + AA), the requirement for IMV (81.6% vs. 79.4%, respectively) does not show statistically significant differences (*p* = 0.877).

## 3. Discussion

The main risk factors for COVID-19 include the male sex and older age [17]. Differences in susceptibility and severity have been associated with genetic polymorphisms within the genes coding cytokines and other pro-inflammatory mediators, increasing the evidence for genetic susceptibility to infection, severity, and outcome of the subjects infected by SARS-CoV-2 [18].

To our knowledge, this is the first study evaluating single nucleotide variants (SNVs) in genes that encode members of the disintegrin and metalloprotease domain or are related to cell adhesion and motility in the lung [14]. We do not find statistical differences in the allele or genotype frequencies for *ADAM33* or *CHRNA5*. The A allele of rs16969968 (*CHRNA5*) was found to be more frequent in smokers than non-smokers; however, this difference was not statistically significant.

The rs16969968 is a non-synonymous variant (G/A) that originates an amino acid change from aspartic acid to asparagine at the 398th residue (D398N) of the acetylcholine receptor subunit alpha-5 protein. The lung of healthy subjects with the AA genotype has 2.5-fold lower protein than patients with the GG genotype [19]. *CHRNA5* is principally expressed in epithelial cells; pentamers (α3β2)2α5 are involved in cell migration and the modulation of calcium influx during wound healing of the respiratory epithelium [20]. At the protein level (α5 subunits), the A allele (rs16969968) has been associated with structural and functional alterations of airway epithelium, contributing to impairment of the epithelial-related immune response of the airway (Appendix A) [11].

Through in vivo, ex vivo, and in vitro approaches, the contribution of rs16969968 in airway epithelial remodeling and the development of emphysema in murine models has been demonstrated by inducing molecular and cellular changes and promoting the inflammatory response. In addition, rs16969968 nasal polyps were more inflamed and presented secretory cell hyperplasia than wild-type carriers [11,21]. Moreover, the considerable upregulation of detected inflammatory mediators highlights a global dysregulation of the immune response [11,22]. Additionally, the role of the α5 subunit in migrating normal airway epithelium cells and tumor invasion in lung cancers [23] have been described. This subunit is expressed by basal cells in the epithelium [24], which are considered to act as progenitor cells [25]. In addition, the expression and the localization of the α5 subunit in airway epithelial cells from bronchi and bronchioles in non-smokers have been reported [26].

A meta-analysis of nine genetic variants with two respiratory diseases (COPD and lung cancer) identified eight polymorphisms significantly related to changes in disease susceptibility. Strong evidence was assigned to six variants, including *CHRNA5* rs16969968 with COPD and lung cancer risk [27]. The genotype (AA) was associated with increased risk and earlier diagnosis of lung cancer, but the beneficial effects of smoking cessation were very similar in those with and without the risk genotype [28]. The normal lung tissue of carriers of the AA genotype shows 2.5-fold lower mRNA levels of *CHRNA5* in comparison to carriers of the GG genotype [19]; maybe this influence directly with the cell differentiation in respiratory epithelia [11] and is probably a risk factor for the development of pulmonary and infectious diseases.

Data from ENCODE and other public databases showed that SNPs with strong evidence might be in presumptive functional regions [27].

We classified the population into GG carriers (homozygous to the common allele) and compared it with patients with GA or AA genotypes. Almost 20% of the participants have one or two copies of the allele of risk (A). Subjects in the GA + AA group have reduced PaO_2_/FiO_2_ compared to GG patients, but this difference was a tendency (*p* = 0.055).

At hospital admission, patients with GA or AA genotype showed high ESR compared to GG carriers; in COVID-19, high ESR values have been associated with severe pneumonia, oxygen requirements, and intensive care needs. In a Brazilian COVID-19 cohort, the levels of ESR were higher in the no-survival, requiring intensive unit care; ESR correlates with WBC, lymphocytes, and CRP [29]. The ESR is an acute-phase reactant increased in severe or critical COVID-19 [30]. According to our data, survival was independent of genotype in rs16969968. In the present study, variables related to coagulation or acute phase reactants were similar in COVID-19 patients.

Patients with COVID-19 and smokers showed a significant association between HS-COV and LS-COV. The presence of allele A and genotype AA (OR = 1.87, CI 1.05–3.34 and OR = 6.05, CI 1.40–25.88, respectively) were associated with a risk of high TI; probably, this association corresponds to nicotine addiction. A systematic review reported that in COVID-19 patients, those ever-smokers had 30–50% more risk of a severe form of the disease than never-smokers [31].

Spearman’s analysis showed exclusively in smokers patients GA + AA a high and positive correlation between fibrinogen and C-reactive protein at hospital admission. Our data are similar to those reported in the Caucasian population with COVID-19. These patients have high fibrinogen levels, which correlate with excessive inflammation and disease severity [32]. The fibrinogen-to-albumin ratio level has been proposed as a marker in coagulopathy and a good indicator of COVID-19 progression [33].

Our study is not exempt from limitations; maybe the main was the inclusion of only severe and critical cases since our center is a third-level hospital. In addition, we did not have albumin data to evaluate the disease prognosis o severity; however, this is the first study that assessed the variant rs16969968 (*CHRNA5*) in severe COVID-19. We did not include a control group; however, previously to the COVID-19 pandemic, we reported in 606 healthy volunteers the genotypic frequency of rs16969968; the AA genotype had 4.1%, surprisingly in smokers (≥20 cigarettes per day) of the same ancestral contribution, the frequency was 6.1%, similar to the frequency of this genotype in the patients with severe COVID-19 [34].

This manuscript contains relevant information that may lead to new research on lung involvement in patients who smoke and have severe COVID-19; at the genetic level, it is pertinent to investigate whether carriers of the risk genotype (AA), exist affectation in cell differentiation in the respiratory epithelia that contributed to severe COVID-19.

## 4. Materials and Methods

We included 917 COVID-19 patients hospitalized at the Instituto Nacional de Enfermedades Respiratorias Ismael Cosío Villegas (INER) in Mexico from August 2020 to December 2021. The SARS-CoV-2 infection was tested through the nasopharyngeal swab PCR test. All patients were adults with critical disease and oxygenation impairment. The participants were invited to participate and signed an informed consent corresponding to this research protocol previously approved for the Ethics Committee in Investigation of INER (approbation code: C53-20). They were divided into two groups, tobacco-smoking patients (*n* = 257) and non-smokers (*n* = 660). The genotype and allele frequencies of two single nucleotide variants, the rs16969968 (*CHRNA5*) and rs3918396 (*ADAM33*), were evaluated in both groups. Subsequently, the variant rs16969968, which has been associated in several populations with respiratory diseases, was assessed depending on their genotype. We collected several laboratory tests from the clinical record, including complete blood count, blood chemistry, and coagulation tests at admission and 7–10 days in the hospital. Finally, we recollected data for smokers’ patients to obtain the tobacco index.

Following informed consent from each patient, 4 mL of peripheral blood in EDTA tubes as an anticoagulant was obtained; subsequently, the DNA was extracted using the kit BDtractTM (Maxim Biotech Inc., San Francisco, CA, USA). The genetic material was quantified in a Nanodrop 2000 device (Thermo Scientific, Wilmington, DE, USA). A sample of optimum purity was considered when the 260/280 ratio was 1.8–2.0. The alleles of each polymorphism were identified for real-time PCR (qPCR) using the StepOnePlus equipment (Applied Biosystems, Foster City, CA, USA) in the modality of allelic discrimination through TaqMan probes, predesigned for the rs16969968 (C__26000428_20) and rs3918396 (C___1276547_20) (Applied Biosystems, Foster City, CA, USA). We used one cycle of pre-PCR at 60 °C, 30 s, and 95 °C, 10 min; the PCR stage had 40 cycles for 15 s at 95 °C and one minute at 60 °C; finally, the post-PCR stage was to 60 °C for 30 s. Five non-template controls were included for each plate.

We applied Mann–Whitney U tests for demographic and clinical numeric variables, and categorical variables were analyzed with χ^2^ with Yates’ correction. All calculations were done with SPSS version 21.0 software (IBM, New York, NY, USA) and EPIDAT version 3.1 (Junta de Galicia and Pan American Health Organization, 2006, Santiago de Compostela, España).

The association analysis of genotypes and alleles was made using 2 × 2 or 2 × 3 contingency tables. We used a genetic model in the comparison by genotype.

Spearman’s correlation was used to assess a possible linear association between two continuous variables; we only considered high or very high correlations (correlation coefficient ≥ ±0.7). The Kaplan–Meier method evaluated the patients’ survival in GG and GA + AA genetic models. We considered it statically significant if *p* < 0.05. We make PCA in variables not correlated at admission at the hospital—Data Availability in ClinVar accession numbers SCV002819136–SCV002819137.

## 5. Conclusions

COVID-19 patients and smokers’ carriers of one or two copies of the risk allele (rs16969968/A) in *CHRNA5* have high ESR and a positive correlation between fibrinogen and C-reactive protein. The variant in ADAM33 (rs3918396) does not show a significative association.

## Figures and Tables

**Figure 1 ijms-24-09811-f001:**
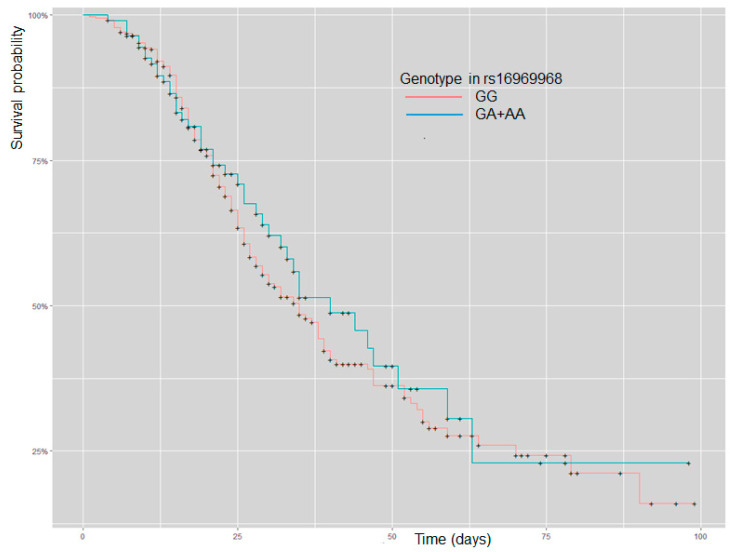
Kaplan–Meier curve analysis for patients with severe COVID-19 and stratification by GG and GA + AA genotype in rs16969968 (*CHRNA5*).

**Figure 2 ijms-24-09811-f002:**
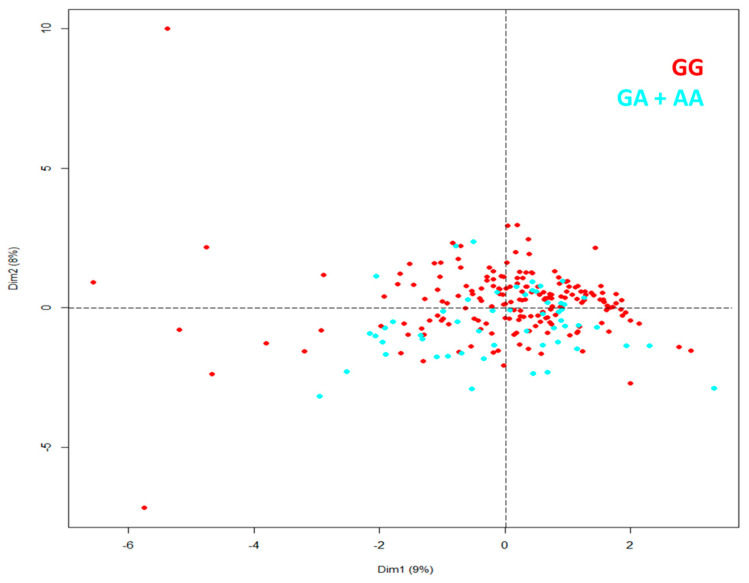
PCA for patients with severe COVID-19 identified by genotype in rs16969968 (*CHRNA5*).

**Table 1 ijms-24-09811-t001:** Genotype and allele analysis of rs16969968 (*CHRNA5*) and rs3918396 (*ADAM33*).

**rs16969968/** ** *CHRNA5* **	**Smokers** **(*n* = 257)**	**Non-Smokers** **(*n* = 660)**	***p*-Value ***
GG	200 (77.8)	537 (81.4)	0.089
GA	49 (19.0)	116 (17.6)	
AA	8 (3.2)	7 (1.0)	
G	449 (87.4)	1190 (90.2)	0.096
A	65 (12.6)	130 (9.8)	
**rs3918396/** ** *ADAM33* **	**Smokers** **(*n* = 257)**	**Non-Smokers** **(*n* = 654)**	** *p* ** **-Value ***
CC	243 (94.6)	612 (93.6)	0.416
CT	14 (5.4)	38 (5.8)	
TT	0 (0)	4 (0.6)	
C	500 (97.3)	1262 (96.5)	0.479
T	14 (2.7)	46 (3.5)	

* *p*-value obtained using $\chi^{2}$ test. The numbers in parentheses are percentages.

**Table 2 ijms-24-09811-t002:** Demographic and clinical characteristics by genotype in rs16969968.

Variable	GA + AA (*n* = 180)	GG (*n* = 737)	*p*
Men (*n*, %) *	119 (66.1)	522 (70.8)	0.251
Age (years) **	61 (52–68)	58 (50–68)	0.267
Days of hospitalization **	19 (13–33)	20 (12–30)	0.807
BMI (kg/m^2^) **	29.4 (26.1–33.3)	29.5 (26.1–33.4)	0.924
Days since symptoms onset **	9 (7–13)	10 (7–14)	0.522
IMV requirement (*n*, %) *	132 (73.3)	578 (78.4)	0.172
IMV (days) **	13 (0–29)	15 (5–25)	0.880
PaO_2_/FiO_2_ (mmHg) **	105 (73–173)	120 (77–189)	0.055
Deaths (*n*, %) *	66 (36.7)	295 (40.0)	0.458
Treated with corticosteroids (%)	84.9	78.5	0.620
Comorbidities (*n*, %) *			
DM type 2	44 (24.4)	215 (29.2)	0.241
Smoking	57 (31.7)	200 (27.1)	0.262
Hypertension	71 (39.4)	244 (33.1)	0.129
Obesity	78 (43.3)	340 (46.1)	0.553
Chronic Lung Disease	19 (10.6)	61 (8.3)	0.410
Ischemic Heart Disease	11 (6.1)	25 (3.4)	0.141

* χ^2^ with Yates’ correction ** U-Mann–Whitney test. Mean ± SD for age, other variables medium (P25–P75). BMI: Body Mass Index. IMV: Invasive Mechanical Ventilation.

**Table 3 ijms-24-09811-t003:** Symptoms according to the rs16969968 (*CHRNA5*).

Variable, *n* (%)	GA + AA (*n* = 180)	GG (*n* = 737)	*p*-Value *
Fever	125 (69.4)	526 (71.4)	0.675
Headache	83 (46.1)	316 (42.9)	0.483
Arthralgias	111 (61.7)	433 (58.8)	0.529
Myalgias	120 (66.7)	449 (60.9)	0.180
Dyspnea	152 (84.4)	624 (84.7)	0.967
Cough	112 (62.2)	510 (69.2)	0.087
Sore throat	39 (21.7)	188 (25.5)	0.329
Rhinorrhea	29 (16.1)	98 (13.3)	0.390
Ageusia	20 (11.1)	112 (15.2)	0.200
Anosmia	5 (2.8)	52 (7.1)	0.050
Thoracic pain	8 (4.4)	69 (9.4)	0.047
Vomit	5 (2.8)	18 (2.4)	0.993
Diarrhea	22 (12.2)	70 (9.5)	0.341

** p*-value was calculated with Yates correction. The numbers in parentheses are percentages.

**Table 4 ijms-24-09811-t004:** Clinical laboratory tests at admission to the hospital.

Variable	GA + AA	GG	*p*
WBC (×10^3^/μL)	9.8 (8.3–13.2)	10.4 (8.0–13.5)	0.505
Lymphocytes (×10^3^/μL)	2 (0.7–6.3)	2.1 (0.7–6.9)	0.818
Platelets (×10^3^/μL)	261 (213–327)	258 (189–334)	0.643
LDH (UI/L)	393 (290–490)	387 (301–508)	0.810
D-Dimer (μg/mL)	1.41 (0.77–3.26)	1.67 (0.60–3.77)	0.814
ESR (mm/h)	32 (23–40)	26 (10–35)	0.038
CRP (mg/dL)	12.0 (5.8–19.5)	12.3 (6.6–20.9)	0.726
Fibrinogen (mg/dL)	707 (575–781)	666 (569–778)	0.493
Procalcitonin (ng/mL)	0.24 (0.08–0.72)	0.21 (0.09–0.65)	0.941
Ferritin (ng/mL)	990 (669–2368)	1064 (637–1892)	0.518

Showing medians and interquartile ranges (p25–p75). The *p*-value was obtained using the Mann–Whitney U test. WBC: white blood cells. LDH: lactic dehydrogenase. ESR: erythrocyte sedimentation rate. CRP: C-reactive protein.

**Table 5 ijms-24-09811-t005:** Genotype and allele analysis of rs16969968 (*CHRNA5*) in smokers patients with severe COVID-19.

rs16969968	HS-COV (*n* = 56)	LS-COV (*n* = 178)	*p* *	OR (CI, 95%)
GG	39 (69.6)	141 (79.2)	0.034	reference
GA	13 (23.2)	35 (19.6)		1.34 (0. 64–2.78)
AA	4 (7.1)	2 (1.1)		7.23 (1.27–40.95)
G	91 (81.2)	317 (89.0)	0.046	
A	21 (18.7)	39 (10.9)		1.87 (1.05–3.34)
GG + GA	42 (92.8)	176 (98.8)	0.030	
AA	4 (7.1)	2 (1.1)		6.05 (1.40–25.88)

* *p*-value obtained using $\chi^{2}$ test. The numbers in parentheses are percentages.

## Data Availability

Data Availability in ClinVar accession numbers SCV002819136–SCV002819137.

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
