# Peer review of "The rs16969968 Tobacco Smoking-Related Single-Nucleotide Variant Is Associated with Clinical Markers in Patients with Severe COVID-19"

_ijms, 2023, doi:10.3390/ijms24129811_

Round 1
Reviewer 1 Report
The authors evaluated the role of two RS16969968 polymorphisms (CHRNA5) and rs3918396 (ADAM33) in patients who developed severe COVID-19.
The article is interesting and provides new insights.
However, it is not clear from the tables whether a statistical analysis has been performed evaluating the correlation between rs3918396 (ADAM33) and clinical features and outcomes.
In addition, the authors should better discuss the possible clinical advantages of knowing the expression of these polymorphisms thus stratifying patients by risk of disease pregression. Since patients were enrolled until December 2021, how many patients were treated with intravenous antivirals or corticosteroids? Did you find differences between those recieving antivirals in the outcomes and in the expressions of these polymorphisms?
Average quality of english
Author Response
The authors evaluated the role of two rs16969968 polymorphisms (CHRNA5) and rs3918396 (ADAM33) in patients who developed severe COVID-19. The article is interesting and provides new insights.
However, it is not clear from the tables whether a statistical analysis has been performed evaluating the correlation between rs3918396 (ADAM33) and clinical features and outcomes.
Thank you for the observation. We added the sentence in lines 77-78 “We made this analysis with rs3918396 (ADAM33); however, it did not show a significant association”.
In addition, the authors should better discuss the possible clinical advantages of knowing the expression of these polymorphisms thus stratifying patients by risk of disease progression.
Thank you for the observation. Regardless of the rs16969968 polymorphism (GG vs. GA + AA), the requirement for IMV (81.6% vs. 79.4%, respectively) does not show statistically significant differences (p=0.877) (lines 140-142), and survival probability is the exact (figure 1). We add possible clinical advantages in the discussion section (lines 181-185 and 221-224).
Since patients were enrolled until December 2021, how many patients were treated with intravenous antivirals or corticosteroids? Did you find differences between those receiving antivirals in the outcomes and in the expressions of these polymorphisms?
We add the data of patients treated with corticosteroids; no significant differences exist (table 2).

Reviewer 2 Report
Article: The rs16969968 tobacco smoking-related single-nucleotide variant is associated with clinical markers in patients with severe 3 COVID-19
Authors present a study where hospitalized COVID-19 patients display a positive correlation between patients carrying a single nucleotide variant within CHRNA5 and several blood tests (erythrocyte sedimentation rate, fibrinogen and C-reactive protein). Authors also mention assessing a single nucleotide variant within ADAM33, but no results beyond table 1 for this SNV are presented and one wonders why this is mentioned at all?
The main issue with this study is no analysis of normal healthy individuals has been conducted. So, study lacks a true control population.
Comments
Line 21: "At admission to the hospital, we collected several laboratory tests from the clinical record"
Comment: this doesn't really say anything for reader. Better explain, or remove.
Line 28: abstract doesn't mention rs3918396 results, only rs16969968. Were rs3918396 polymorphisms not significant? please state in abstract.
Line 42: "ACE2" please define acronym before use.
Line 44: "many factors are involved,"
Question: involved in what?
Line 49: "A single nucleotide variant (SNV) rs16969968 (A allele) has been associated with..."
Question: I'm not understanding what authors mean here. Could authors provide (in supplementary material) where in gene sequence this SNV occurs? Results suggest in can be a single or double G to A variant? is this true?
Line 218: "Subsequently, the variant rs16969968, which has been associated in several populations with respiratory diseases, was assessed depending on their genotype."
Question: why was only rs16969968 assessed? Why include rs3918396 in this study at all?
Line 247: Authors mention rs16969968 in conclusion, but no mention of rs3918396. If rs3918396 was not correlated with anything this should also be mentioned.
Suggestions
Lines 16, 60: suggest change "y" to "and"
Line 32: suggest change "ranging from asymptomatic people to developing acute respiratory distress syndrome" to "ranging from some individuals being asymptomatic to others developing acute respiratory distress syndrome..."
Line 66: please state in table 1 legend that number in () is percent?
Line 223: suggest change "Previous" to "Following"
Over all English is good and very understandable.
there were a couple of situations where "y" was used instead of "and" (see comments above)
Author Response
We appreciated the observations and comments of the reviewer.

Round 2
Reviewer 2 Report
I thank the authors for their edits.
While I still have some doubt regarding the over-all novelty of this study, I'm satisfied there is benefit in its publication.